# 3D ink-extrusion additive manufacturing of CoCrFeNi high-entropy alloy micro-lattices

Christoph Kenel [1], Nicola P.M. Casati [2] & David C. Dunand [1]

Additive manufacturing of high-entropy alloys combines the mechanical properties of this novel family of alloys with the geometrical freedom and complexity required by modern designs. Here, a non-beam approach to additive manufacturing of high-entropy alloys is developed based on 3D extrusion of inks containing a blend of oxide nanopowders ($Co_3O_4 + Cr_2O_3 + Fe_2O_3 + NiO$), followed by co-reduction to metals, inter-diffusion and sintering to near-full density CoCrFeNi in $H_2$. A complex phase evolution path is observed by in-situ X-ray diffraction in extruded filaments when the oxide phases undergo reduction and the resulting metals inter-diffuse, ultimately forming face-centered-cubic equiatomic CoCrFeNi alloy. Linked to the phase evolution is a complex structural evolution, from loosely packed oxide particles in the green body to fully-annealed, metallic CoCrFeNi with 99.6 ± 0.1% relative density. CoCrFeNi micro-lattices are created with strut diameters as low as 100 μm and excellent mechanical properties at ambient and cryogenic temperatures.

[1] Department of Materials Science and Engineering, McCormick School of Engineering, Northwestern University, Evanston, IL 60208, USA. [2] Swiss Light Source, Paul Scherrer Institut, 5232 Villigen, Switzerland. Correspondence and requests for materials should be addressed to C.K. (email: Christoph. Kenel@northwestern.edu)

Fabrication of engineering parts from high-entropy alloys (HEA) using additive manufacturing (AM) has been increasingly studied in the past five years[1–3]. Initial studies focused on powder bed fusion of CoCrFeNi, a prototype single-phase high entropy alloy with face-centered-cubic (fcc) structure[4], from pre-alloyed powders, and observed increased strength and ductility compared to cast alloys[3]. Later, the Al-Co-Cr-Fe-Ni system was predominantly studied, either as equiatomic AlCoCrFeNi or as $Al_xCoCrFeNi$ with $0.1 < x < 1$[5–9]. Other work demonstrated AM of CoCrFeMnNi by selective laser melting[10], $Co_{1.5}CrFeNi_{1.5}Ti_{0.5}Mo_{0.1}$ by directed energy deposition[5], ZrTiVCrFeNi by electron beam melting[11], and polymer-AlCoCrFeNi composites by coating 260 nm thick 3D printed polymer nanolattices with 14–126 nm thick HEA shells[12]. It is typically observed that AM processing through the liquid state, followed by solidification at high cooling rates, refines the grain size, leads to finer precipitates in two-phase alloys, and produces a highly textured microstructure; however, it may require high preheating temperatures to attenuate cracking, and it leads to inter-dendritic segregation of elements and microstructural gradients due to the imposed thermal history in the layer-wise process[5]. Apart from the currently dominating laser- and electron-beam based AM methods, alternative methods to metal AM have been developed, such as binder jetting[13–17] and 3D ink-extrusion printing[18] where first a binder-containing green body is shaped at ambient temperature from elemental or pre-alloyed powders that is then densified in a subsequent isothermal sintering step. These techniques have the potential to provide segregation-free, structurally-homogeneous alloys with low residual stress due to full inter-diffusion and isothermal sintering, while also reducing cost and eliminating the need of inert-gas processing environments of beam-based AM. The inks for 3D ink-extrusion can be produced from alloyed metal powder, elemental powder or compounds that reduce (e.g., oxides) or decompose (e.g., hydrides) to metal upon thermal processing[18–27]. The elimination of pre-alloying steps (i.e., to create powders for selective laser melting) and the ability to directly produce alloys from blended pure oxide feedstock further reduce cost, time, and provide full flexibility of alloy composition. The use of oxides is limited to those which can be reduced with gases, such as CO or $H_2$. Under hydrogen, pressed $Cr_2O_3$ pellets require 18 h at 1373 K for full reduction, making this approach unpractical for commercial production of pure Cr metal[28]. For alloyed systems, sequential reduction was observed upon co-reduction of $Fe_2O_3 +$ NiO[29] and $Co_3O_4 +$ NiO[30] powder blends, with metallic Ni forming first. Synergistic effects have been found in co-reduced $FeCr_2O_4$[31,32] (or $CoCr_2O_4$[33]) powders, where first a Fe (or Co-) matrix, is formed which then acts as an acceptor for reduced Cr atoms creating a Fe-Cr (or Co-Cr) alloy. Co-reduction of blended $Fe_2O_3 +$ NiO $+ Cr_2O_3$ powders was found to proceed as a combination of the binary sub-systems: metallic Ni forms first, then forming Ni-Fe solid solutions and gradually incorporating Cr until a Ni-Fe-Cr alloy is achieved[34]. Using this approach, complex alloys, such as martensitic and maraging steels, can be produced from blended oxide precursors[35–37]: for example, honeycomb Fe-Ni and Fe-Cr structures have been manufactured via extrusion of blended $Fe_2O_3 +$ NiO and $Fe_2O_3 + Cr_2O_3$ slurries followed by reduction in $H_2$[37,38].

In this work, we demonstrate an approach to AM of HEAs by 3D extrusion printing of inks containing a blend of $Co_3O_4 + Cr_2O_3 + Fe_2O_3 +$ NiO nanometric powders, followed by co-reduction and sintering to yield equiatomic Co-Cr-Fe-Ni, a prototype alloy for single-phase fcc high entropy alloys[4]. We study the phase and microstructural evolution throughout thermal processing, from extrusion-printed oxide to fully-densified HEA filaments. A fcc CoCrFeNi HEA is obtained with near-full density ($0.4 \pm 0.1\%$ porosity) and a minimal feature size (filament diameter) of 100 μm. In situ X-ray diffraction, together with thermogravimetry, reveal the kinetics of reduction and inter-diffusion upon thermal processing, starting from loosely-packed, as-printed oxide powders to freshly-reduced metallic particles, with sub-micrometer size which is crucial in rapidly achieving near-full densities. Mechanical testing of sintered single filaments and micro-lattices show an excellent combination of ductility and strength at ambient and cryogenic temperatures.

## Results and discussion

### 3D ink-extrusion, reduction, and sintering of CoCrFeNi HEA.
The ink used for 3D ink-extrusion consists of a blend of $Co_3O_4$, $Cr_2O_3$, $Fe_2O_3$, and NiO powders, poly-lactic-co-glycolic-acid as binder, dibutyl phthalate as plasticizer, and ethylene glycol butyl ether as a surfactant. Thermogravimetric analysis (TGA) in a $H_2$ atmosphere of the blended oxide powders without binder shows reduction in two major steps (Fig. 1). First, the metals Co, Fe, and Ni are formed in rapid succession, leading to a mass loss of 20.5%, close to the theoretical stoichiometric value of 19.7%. In a second step—and with a roughly tenfold slower mass loss rate—the more stable $Cr_2O_3$ is reduced to Cr, which diffuses into the adjacent Co-Fe-Ni matrix, increasing its Cr content. The final mass loss after holding under $H_2$ for 1 h at 1557 K is measured at 28.3%, in good agreement with the theoretical value of 27.5%. The oxide powders, blended using wet mill-mixing, show <10 μm agglomerates and a <1 μm grain size, enabling a large surface area and the observed fast initial reaction with $H_2$ (Fig. 1b). The small agglomerate size is also crucial for 3D extrusion printing with fine nozzles without clogging. X-ray diffraction of the blended oxide powder prior and after the TGA measurement demonstrates the structural change from blended oxides to a single-phase fcc CoCrFeNi HEA with no oxide diffraction rings left (Fig. 1c). This clearly illustrates the feasibility of creating HEA from blended oxide powder feedstock and provides the base for 3D ink-extrusion of complex HEA objects. The ink for 3D ink-extrusion is mill-mixed and thickened for the shape-defining first step (Fig. 1d, top). This defines the architecture and geometrical relations of the part. In a second step, the blended oxide is converted to metallic CoCrFeNi HEA by co-reduction, inter-diffusion and sintering in $H_2$ atmosphere. Despite the drastic changes in material properties from an oxide-polymer composite to a fully-annealed, densified, metallic CoCrFeNi HEA and the associated mass (−40%) and volume changes (−78%), the architecture of the micro-lattice is fully conserved without warping or cracking, due to isotropic and homogeneous shrinkage (Fig. 1d, bottom). The extensive but uniform shrinkage has the advantage to allow the creation of features (i.e., struts) with diameters below that of the extrusion nozzle used. In the example shown here, printing was performed with a 200 μm diameter nozzle, resulting in a HEA strut diameter of 103 μm.

### Phase evolution by in situ synchrotron X-ray diffraction.
A detailed analysis of the phase evolution of 250 μm diameter extruded filaments including the binder during co-reduction is performed using in situ synchrotron X-ray diffraction (XRD) in flowing $H_2$ (Fig. 2). The as-printed filament shows only the expected peaks for the $Co_3O_4$, $Cr_2O_3$, $Fe_2O_3$, and NiO powder (Fig. 2a). After in situ co-reduction, a fcc CoCrFeNi HEA is observed, with a lattice constant of 3.575(±0.002) and 3.6468 (±0.0004) Å at 293 and 1215 K, respectively. The average coefficient of thermal expansion, calculated as $21.6 \pm 0.5 \times 10^{-6}$ K$^{-1}$, is in general agreement with the scarce literature values available from linear expansion measurements of bulk specimens[39,40]. Detailed peak analysis reveals the presence of two fcc phases with

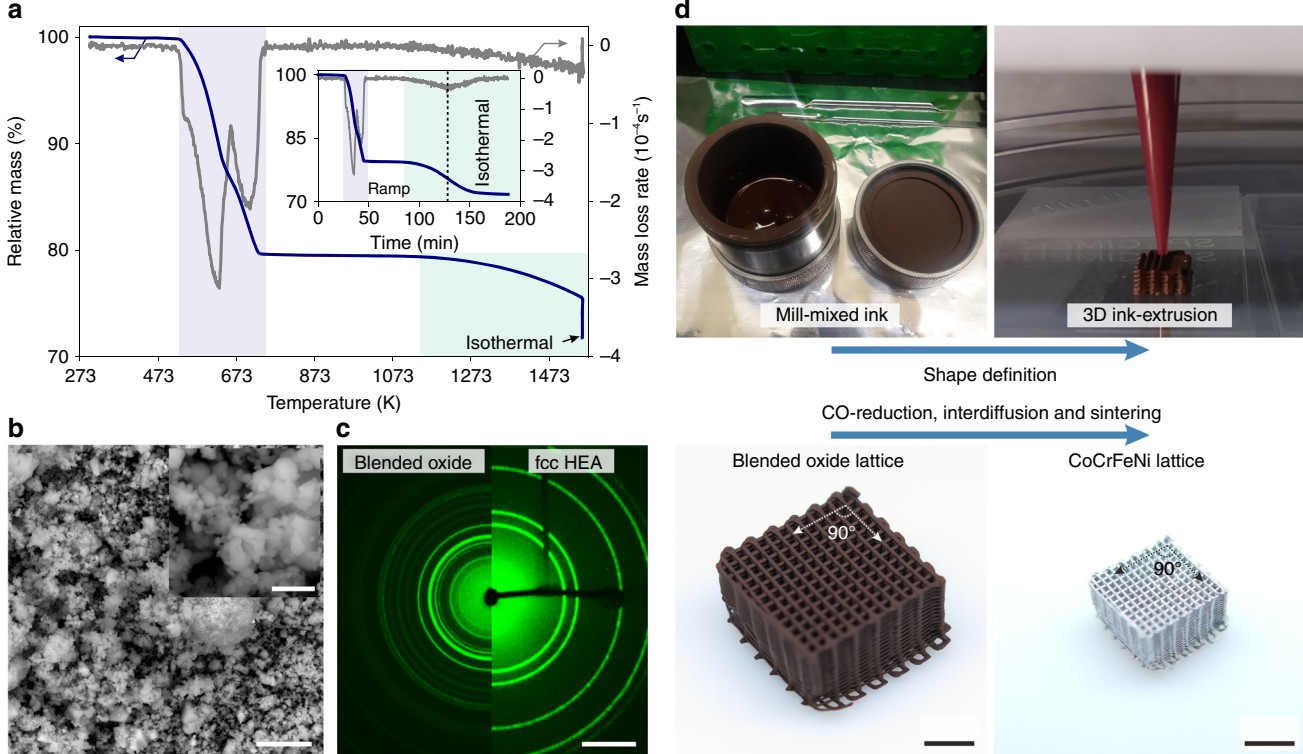

**Fig. 1** Creating CoCrFeNi high-entropy alloys via 3D ink-extrusion, reduction and sintering of oxide powder blends. **a** TGA mass loss of a loose blend of $Co_3O_4 + Cr_2O_3 + Fe_2O_3 + NiO$ as function of temperature in a $H_2$ atmosphere. Initial reduction of $Co_3O_4$, $Fe_2O_3$ and NiO (blue shading) is followed by reduction of $Cr_2O_3$ (green shading). Inset: Same data as function of time, showing ramp and 1 h hold at 1557 K, when continuous mass loss occurs due to ongoing reduction of $Cr_2O_3$ until it is fully consumed. **b** Scanning electron micrograph of the blended oxide powder prior to TGA, showing agglomerate sizes <10 μm and sub-micron particle size. Scale bars are 5 μm and 1 μm (inset). **c** Combined 2D diffractograms of blended oxide prior (left) and after (right) the TGA measurement in $H_2$ showing the complete disappearance of all oxide-related diffraction intensities and formation of a face-centered cubic CoCrFeNi HEA. Scale bar is 20 nm⁻¹. **d** The two-step processing route first shapes the mill-mixed ink using 3D ink-extrusion into the desired architecture (a micro-lattice with 200 μm diameter struts, in the present example). In a second step, the powders in the green body are co-reduced, inter-diffused and sintered to yield a CoCrFeNi HEA micro-lattice with much smaller overall dimensions and strut diameter, without warping or cracking. Scale bars are 3 mm

a difference of $2.6 \times 10^{-3}$ Å in their lattice spacings (Supplementary Figure 1). While this peak doublet could reflect incomplete inter-diffusion and homogenization, other high resolution X-ray diffraction studies have produced similar results on as-cast and homogenized CoCrFeNi, indicating that many high-entropy alloys might consist of multiple, closely-related structures with the same symmetry[41]. Upon oxide co-reduction, the rapid sequential formation of metallic Co, Ni and Fe is observed while $Cr_2O_3$ persist until its slow reduction above 1073 K (Fig. 2b), in agreement with the TGA results. The faster reduction of $Cr_2O_3$ in the in situ XRD experiment compared to the TGA is explained by the faster $H_2$ flow conditions in the capillary setup and the higher purity of the $H_2$ gas (6 N vs. 5 N), effectively lowering the local $O_2$ partial pressure. The complexity of the phase evolution upon co-reduction is illustrated in Fig. 2c. Initial sequential reduction of NiO to fcc(Ni), of $Fe_2O_3$ to $Fe_3O_4$ to body centered cubic (bcc) Fe, and of $Co_3O_4$ to fine grained hexagonal close packed (hcp) Co with inter-dispersed stable $Cr_2O_3$ occurs within a narrow temperature window, between 573 and 673 K. At higher temperature, the hcp(Co) transforms to fcc(Co) and inter-diffusion leads to the formation of additional fcc(Fe,Ni) and bcc(Co,Fe) phases. With the onset of $Cr_2O_3$ reduction at 1073 K, a fourth fcc phase appears, consuming all other phases and becoming the final fcc CoCrFeNi HEA phase. This phase then remains stable upon cooling back to room temperature, allowing the determination of the above coefficient of thermal expansion (Supplementary Figure 2).

**Densification and mechanical performance of CoCrFeNi HEA.** Microstructure and integrity of additively-manufactured HEA material are crucial for load-bearing applications. The structural evolution upon co-reduction and sintering is studied in detail on single filaments (Fig. 3). In the as-extruded state, the powder particles are loosely packed in the solidified ink and held together by the binder. Upon de-binding and initial reduction of $Co_3O_4$, $Fe_2O_3$, and NiO (to Co, Fe, and Ni, respectively) up to 765 K, this loose particle arrangement is conserved (Fig. 3, top). The presence of unreduced $Cr_2O_3$ reduces the number of contact points between reduced, metallic particles, thus acting as a sintering inhibitor. Larger agglomerates, undergoing local sintering as well as volume reduction of larger oxide particles via chemical reduction, lead to the formation of porous, spongy Co, and Ni. Above 1073 K, fine $Cr_2O_3$ particles reduce to metallic Cr, which then diffuses into, and alloys with, the surrounding porous metal matrix (Supplementary Figure 3). Remaining larger $Cr_2O_3$ particles continue to inhibit densification locally, effectively acting as diffusion barriers. Sintering of the metallic matrix around these micrometer-sized remaining $Cr_2O_3$ particles lead to their encapsulation which strongly hinders access of $H_2$ and removal of $H_2O$ (Fig. 3, center). Final reduction of remaining $Cr_2O_3$ particles and densification to a near-fully dense state is achieved after sintering at 1573 K (Fig. 3, bottom). The obtained microstructure exhibits equiaxed grains, 5 to 25 μm in size, containing annealing twins; this is a microstructure also observed in cast and recrystallized fcc CoCrFe(Mn)Ni HEAs[42]. This similarity in

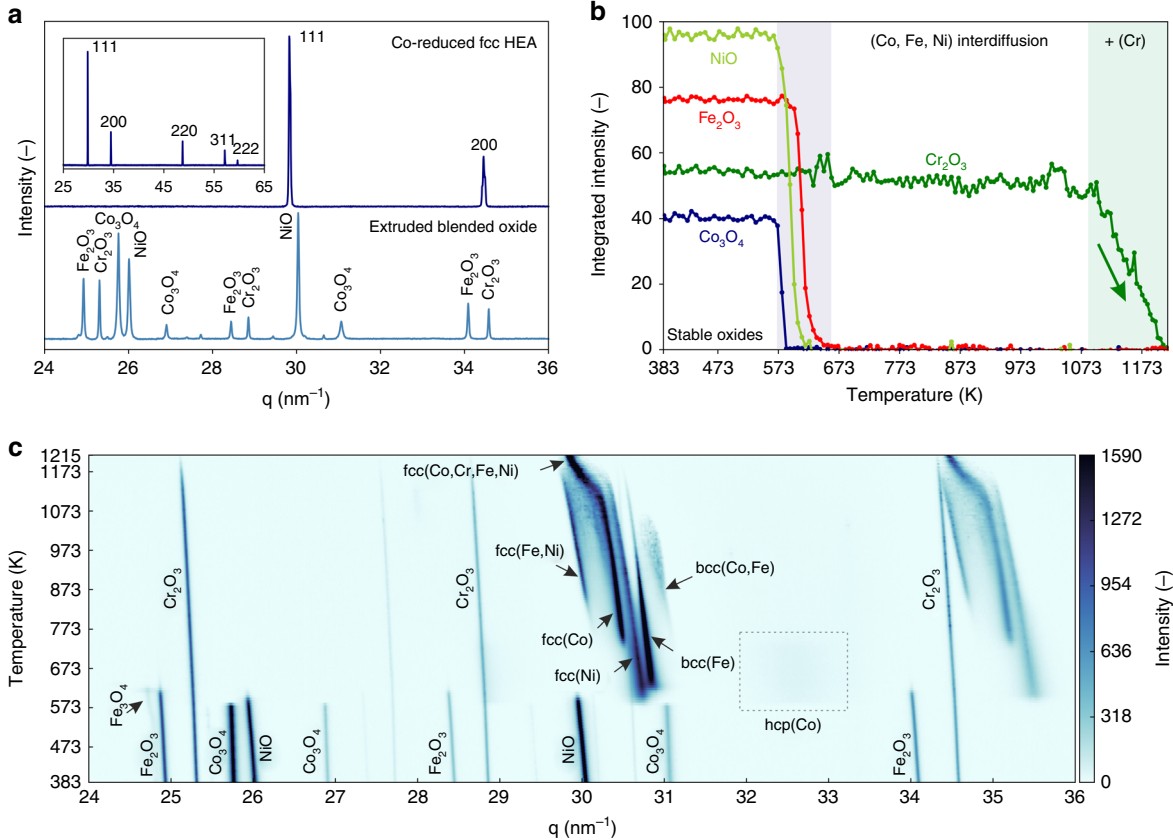

**Fig. 2** Phase evolution upon co-reduction of 3D printed blended oxides by in-situ synchrotron XRD in $H_2$. **a** XRD diffractograms of a 250 μm diameter filament extruded from blended $Co_3O_4 + Cr_2O_3 + Fe_2O_3 + NiO$ inks (bottom) and the inter-diffused fcc HEA filament after completion of the co-reduction step (top). Inset: full data range of the fcc HEA alloy. **b** Evolution of the integrated peak intensities for $Co_3O_4$, $Cr_2O_3$, $Fe_2O_3$, and NiO, demonstrating sequential reduction forming Co, Ni, Fe between 573 and 637 K, followed by reduction of $Cr_2O_3$ between 1073 and 1215 K. **c** 2D phase evolution plot (temperature vs. the scattering vector q, with diffraction peak intensity as color map) during in situ XRD upon heating and reduction in $H_2$, illustrating the complex pathway to CoCrFeNi HEA alloy formation

microstructure allows to apply knowledge in microstructure-property relationships obtained in HEAs via conventional processing routes, combined with the geometrical freedom and complexity offered by AM. Residual sub-micrometer porosity and swelling voids observed for the 3D ink-extruded material are telltale features of the sintering process (Fig. 4a). With increasing sintering temperature, higher densification is achieved for a sintering time of 1 h. No significant difference is observed between filaments extruded from nozzles spanning 200 to 510 μm in diameter, demonstrating the ability of the process to create components with variable wall or strut thicknesses (Supplementary Figure 4). The maximum density achieved is 99.6 ± 0.1 % for sintering at 1573 K for 1 h, averaged over all nozzle diameters. This is substantially higher compared to prior work on pre-alloyed CoCrFeMnNi powders, where 87% was achieved upon pressure-less sintering of loose powder, and 97% for electron-beam melting of powder beds. Only spark plasma sintering yields comparable densities to the sintered HEA of this work[43]. The sintering mechanism of mechanically-alloyed CoCrFeNi with a grain size of <20 μm was found to be dominated by Co and Ni diffusion[44]. Based on the in situ XRD data of our work, Ni and Co are the first metals to appear upon co-reduction and thus initiate densification upon sintering of ink-extruded CoCrFeNi filaments. For flaw-sensitive mechanical applications, hot isostatic pressing could be applied to completely eliminate the small residual closed porosity. The slight iso-thermal swelling phenomenon observed at 1573 K is attributed to diffusion of freshly reduced Cr into the surrounding metallic

matrix, thereby expanding its lattice parameter, as seen by XRD, and the overall volume[45] (Fig. 4b).

Fully sintered filaments (1573 K, 1 h), with a nearly circular cross-section and a grain size of 35–80 μm, an average diameter of 125 μm and a gauge length of 10 mm, were tested in tension until fracture. The stress-strain curves are repeatable, showing a yield strength of 250 ± 5 MPa, extensive strain hardening and an ultimate tensile strength of 598 ± 8 MPa at 33.8 ± 1.3% fracture strain at 293 K (Fig. 4c). At 130 K, the yield stress, ultimate tensile strength and fracture strain are all increased to 388 ± 7 MPa, 864 ± 12 MPa, and 37.6 ± 0.7%, respectively, demonstrating the outstanding combination of strength and ductility of CoCrFeNi at cryogenic temperatures. While yield and ultimate strengths are comparable to, or exceed, properties of cast CoCrFeNi (Supplementary Table 1)[46–48], elongation to fracture is reduced; this is consistent with the small number of grains across the filament diameter (2–3 grains based on a grain size of 35–80 μm), which is an effect also observed for $Al_{0.25}$CoCrFeNi coarse-grained (~100 μm) sheets upon reduction of their thickness[49]. In general, the measured yield stress in this work is expected to be lower compared to the bulk value due to the small number of grains across the filament diameter and thus reduced constraint strengthening[50]. Upon testing of filaments with a reduced gauge length of 1 mm, the stress-strain curves exhibit more variability, which is consistent with oligo-crystalline behavior: serrations and jumps in the stress-strain curves indicate activation and subsequent hardening of slip systems in individual grains with favorable orientations, thus affecting yield stress and ductility.

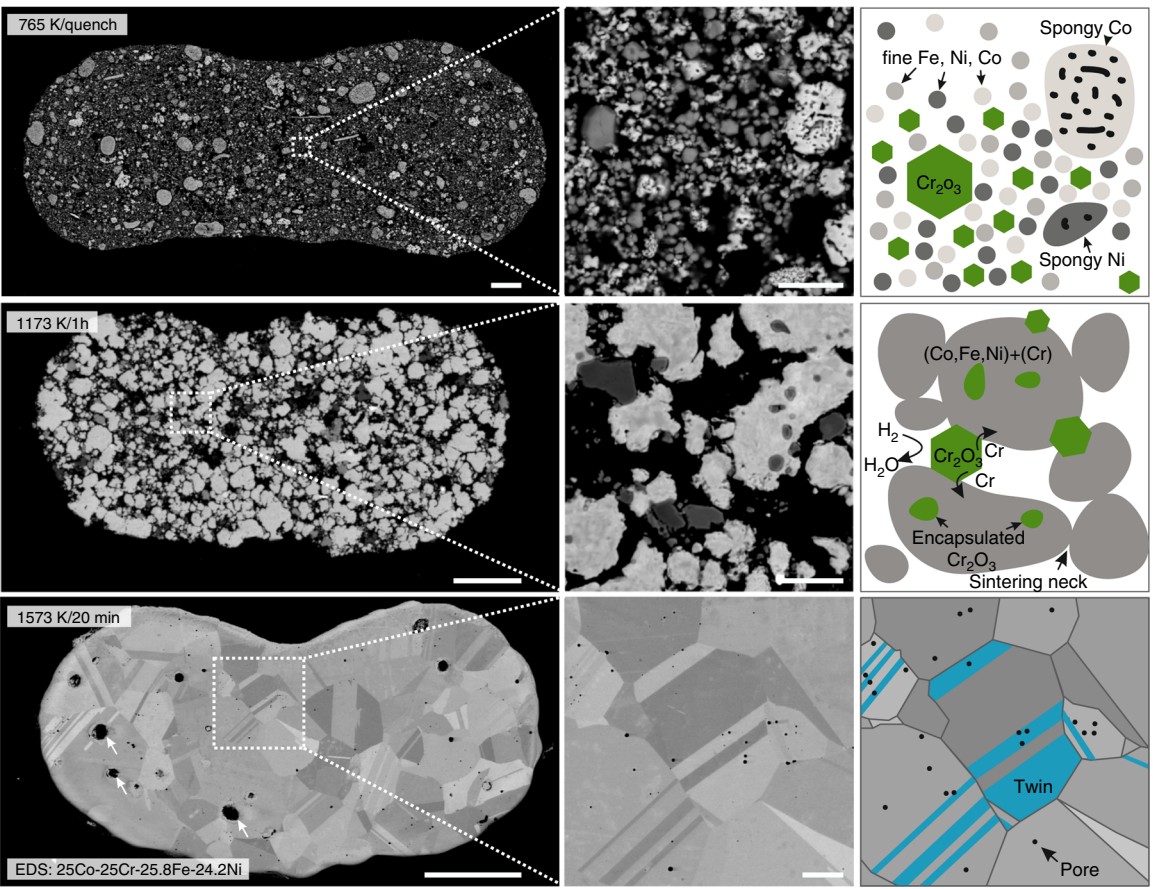

**Fig. 3** Microstructural evolution of 3D extrusion printed CoCrFeNi HEA filaments. top) Cross-section of a 3D extruded filament heated to 765 K in $H_2$ and quenched in Ar. A fine structure of metallic Fe, Ni and Co with inter-dispersed $Cr_2O_3$ is observed (right). center) Filament sintered at 1173 K for 1 h showing sintering of the (Co, Fe, Ni, Cr)-matrix leading to partial encapsulation of $Cr_2O_3$ (right). Smaller $Cr_2O_3$ particles were reduced to Cr and dissolved (Supplementary Figure 3). bottom) CoCrFeNi HEA filament sintered at 1573 K for 20 min showing a coarsened grain structure (grain size of 5 to 25 μm) with annealing twins, swelling voids (~2–5 μm, arrows) and sub-micrometer pores. Filaments are flattened due to printing on a substrate and gravitational sagging. Scale bars are 25 μm for low magnification (left) and 3 μm for high magnification (center) micrographs

Additionally, the reduced gauge length increases the sensitivity for localized effects such as necking before fracture (Supplementary Figure 5). The fracture mode is ductile and dimples on the fracture surface are indicative of ductile void coalescence (Fig. 4c, inset). This result is in contrast to recent observations made for CoCrFeNi tensile specimens charged in 120 MPa $H_2$ gas, where a change in fracture mode to intergranular fracture was observed[51]. However, increased ductility and strength of electrochemically $H_2$-charged CoCrFeMnNi HEA tensile specimens have also been described[52]. No such transition is observed for our CoCrFeNi filaments after sintering at 1573 K for 1 h and furnace cooling in pure $H_2$ at ambient pressure: ductile behavior is prevalent over the complete fracture surface (Supplementary Figure 6). Sub-micrometer oxide precipitates observed in the dimples are measured by EDS to be rich in Si, Ca, Mg, Al, and Na: these elements were present as impurities in the original oxides and were not reduced to metal.

The load-bearing capability of complex-shaped 3D ink-extruded CoCrFeNi specimens is further demonstrated by compression testing at 77 and 293 K of a 0/90° cross-ply micro-lattices with a solid volume fractions of 25% (103 ± 12 μm diameter wide struts with 363 ± 17 μm center-to-center spacing) to 63% (285 ± 38 μm diameter wide struts with 498 ± 23 μm center-to-center spacing) (Fig. 4d). Due to stress concentrations at strut contact points, the micro-lattices have a narrow elastic range, beyond which they undergo stretching-dominated plastic deformation with continuously-increasing stress up to a strain of 50%; at higher strains, the cross-ply stacked structure of struts densifies due to plastic collapse of vertical inter-strut channels. Throughout the compressive deformation, the lattices show no signs of catastrophic cracking or abrupt loss in strength (stress serrations) up to their compaction above 50% strain, demonstrating high energy absorption and an outstanding combination of strength and ductility (Supplementary Movie 1). Direct comparison of two lattices, with an identical relative density of 60%, tested at 77 and 293 K, reveals a higher yield stress and a higher strain hardening rate at 77 K, combined with excellent ductility. This combined effect yields higher energy absorption (area under the stress-strain curve) at 77 K compared to 293 K.

AM of HEAs via directed-energy melting/sintering of pre-alloyed powders is an emerging field; the approach presented here is a low-cost alternative method, based on ambient-temperature ink-printing of inexpensive oxides, followed by hydrogen reduction and sintering, which yields compositionally-homogeneous, dense, strong, ductile HEA filaments or 3D structures. The use of blended powders—oxide as shown here, but which could be also metallic—allows full compositional flexibility well beyond the particular CoCrFeNi composition studied here, and even compositionally-graded HEAs, which can be printed by changing the mixing ratios of multiple inks. This will also allow this

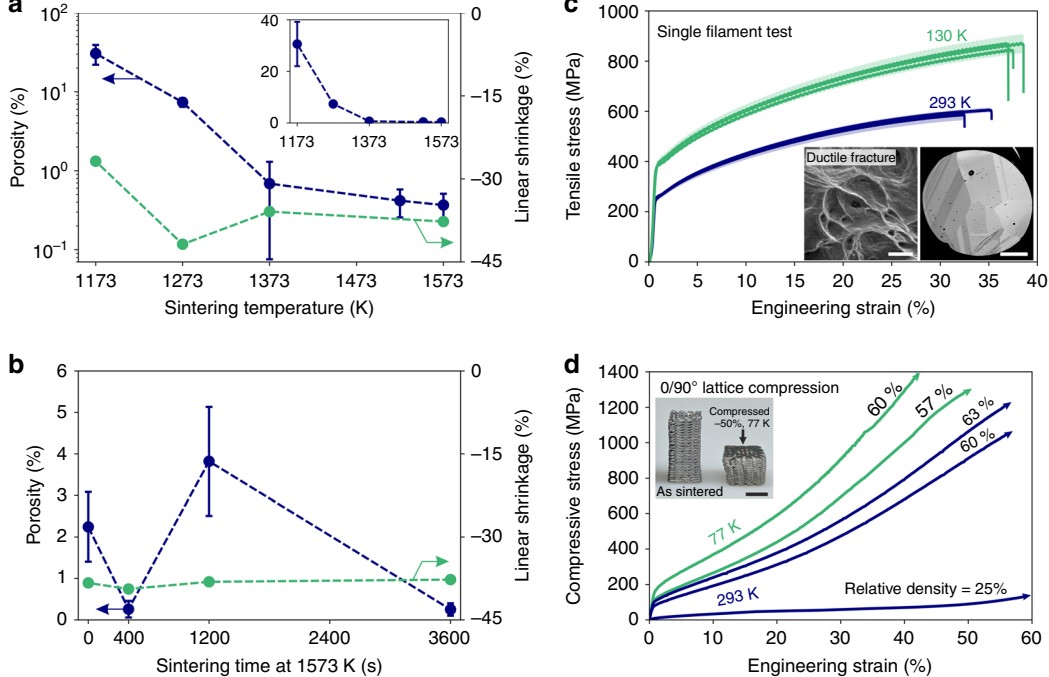

**Fig. 4** Porosity evolution and mechanical performance of extrusion-printed 250 μm CoCrFeNi HEA filaments. **a** Evolution of porosity (logarithmic scale, blue) and shrinkage (linear scale, green) with increasing sintering temperature in $H_2$ for 1 h dwell time. Extensive, reproducible linear shrinkage is observed, with a maximum at 1273 K. Averaged data over all nozzle diameters. Inset: Porosity on a linear scale for comparison. Error bars correspond to ± σ. **b** Evolution of porosity and shrinkage (both linear scale) with sintering time at 1573 K in $H_2$. Error bars correspond to ±σ. **c** Engineering stress-strain curves of single filaments (125 μm average diameter, 10 mm gauge length, from 250 μm nozzle, sintered at 1573 K/1 h) tested in tension at 130 K (green line) and 293 K (blue line) until fracture. Shading corresponds to the 1σ error in calculated stress based on the measured cross-sectional area variation of filaments. Inset: fracture surface showing dimples typical of ductile fracture mode at 293 K (scale bar 1 μm) and cross-section of a filament showing its oligocrystalline structure (grain size: 35–80 μm). Scale bar is 25 μm. **d** Compressive stress-strain curve of 0/90° cross-ply CoCrFeNi HEA micro-lattices with 25% (when extruded with a 200 μm nozzle) to 63% (250 μm nozzle) relative density, sintered at 1573 K for 1 h, and tested at 77 K (green) and 293 K (blue), showing extensive plastic deformation at all temperatures. All tests were interrupted prior to fracture. Inset: Sintered lattice before (left) and after (right) compression to −50% at 77 K, showing homogeneous deformation. Scale bar is 2 mm

technique to be used in alloy development by creating a variety of compositions, in short times, for combinatorial discovery of novel alloys, as demonstrated using beam-based deposition of metal powder blends[53]. Additionally, this method can create many other printed micro-lattice geometries[18] and the 3D ink-extruded structures can be modified in their green body state, as demonstrated by creating origami and kirigami structures from ink-extruded $TiH_2$ sheets[19]. The CoCrFeNi HEA produced here can find direct applications in low-temperature applications requiring high ductility and fracture toughness, based on the unique properties of the CoCrFe(Mn)Ni HEA at cryogenic temperatures[42]. Combined with the geometrical freedom of extrusion printing to create micro-lattices at low cost, this material-process combination could be scaled up for high-volume production of light-weight, high-ductility, impact-absorbing HEA structures than can operate at temperatures ranging from liquid nitrogen up to ~1000 °C, or even beyond, if coatings are used, e.g., via pack aluminization. With recent advances in AM using lithography and direct laser writing techniques, sub-micron scale lattices forming metamaterials[54] and HEA-polymer composites[12] showing superior mechanical properties have been demonstrated. Thus miniaturization of HEA cellular structures provides a potential path to discovery of combinations of structures and alloys further pushing the limits of their mechanical performance beyond the—already impressive—bulk properties of HEAs. The approach presented in this work, co-reduction of blended oxide nanopowder inks, allows to create such complex composites or purely metallic HEA micro- and nano-lattices.

## Methods

**Ink preparation, 3D printing, and thermal treatment**. The extrudable ink (target composition after co-reduction: 25Co-25Cr-25Fe-25Ni, at.%) with 70:30 vol.% powder-to-polymer ratio is prepared from $Fe_2O_3$ (2.57 g, ≥99%,<5 μm, Sigma-Aldrich), NiO (2.40 g, 99%, −325 mesh, Sigma-Aldrich), $Co_3O_4$ (2.58 g, <10 um, Sigma-Aldrich) and $Cr_2O_3$ (2.45 g, 99.7%, −325 mesh, Alfa Aesar) powders, poly-lactic-co-glycolic-acid (PLGA, 0.87 g, 82:18, Evonik Industries) as binder, dibutyl phthalate (DBP, 1.58 g, Sigma-Aldrich) as plasticizer, ethylene glycol butyl ether (EGBE, 0.79 g, Sigma-Aldrich) as surfactant and methylene chloride (20 ml, DCM, Sigma-Aldrich) as solvent. Wet mill-mixing of the oxide powder is performed with 8 ml DCM and EGBE for 30 min (WC-Co, ball-to-powder ratio: 2.15) and then combined with the dissolved PLGA and DBP. The low viscosity ink is thickened by solvent evaporation to a viscosity of ~40 Pa s. Extrusion printing is performed on an Envisiontec 3D Bio-plotter using tapered plastic nozzles (200 and 250 μm, Nordson EFD) and straight metal nozzles (330 and 510 μm, Nordson EFD). Extruded filaments are co-reduced and sintered in flowing $H_2$ (99.999%, dew point: 201 K, 250 ml min$^{-1}$, Airgas) in a commercial $H_2$ tube furnace (10 K min$^{-1}$, GSL-1500-50HG, MTI) in alumina boats. De-binding was performed in two 30 min steps at 150 and 300 °C.

**In situ X-ray diffraction**. In situ X-ray diffraction during heating (10 K min$^{-1}$) in reducing $H_2$ atmosphere (99.9999%, 1.3 ml min$^{-1}$) is performed with a mono-chromatic beam (19.9 keV, 0.7 × 1.4 mm) at the Material Science powder diffraction beamline X04SA[55] of the Swiss Light Source (Paul Scherrer Institut, Switzerland). The as-printed filament segment (2 mm in length and 250 μm in diameter) is contained in a 500 μm outer diameter quartz capillary with 10 μm wall thickness (Hilgenberg, Germany), which is cut and joined on both ends to stainless steel tubing using epoxy adhesive (Araldite Rapid, Huntsman). A gas-tight connection is achieved by a compression fitting (Swagelok, USA) with graphite ferrules. Heating is realized by a hot-air blower (Cyberstar, France) located under the quartz tube (Supplementary Figure 7). The hot blower gas temperature at the specimen position is calibrated prior to diffraction runs over the complete temperature range by placing a thermocouple instead of the specimen. Diffracted

intensities are recorded on a Mythen II array simultaneously covering 120° of 2θ, at an exposure time of 24 s and 3° rocking angle to increase particle statistics. Silicon powder (NIST 640c) is used as calibration standard at room temperature. Data post-processing and plotting are performed using Python (Anaconda, Continuum Analytics). Perceptually uniform color maps are retrieved from the cmocean package[56]. All diffractograms are background-corrected using asymmetric least squares smoothing[57].

**Characterization**. The mass loss of milled feedstock powder without binder addition is measured by thermogravimetry (99.999% $H_2$, 100 ml min$^{-1}$, 10 K min$^{-1}$, $Al_2O_3$ crucible, Netzsch STA 449F5). Tensile testing of reduced, sintered HEA single filaments with an average diameter of 125 μm is performed on a RSA-G2 (TA Instruments, USA) mechanical analyzer with gauge lengths of 1 and 10 mm using a 35 N load cell (Supplementary Figure 8). Compressive testing of a 3D-printed, reduced and sintered HEA lattices (25%: 5 × 5 × 2.6 mm, 200 μm nozzle; 57–63%: 2.75 × 2.75 × 6 mm, 250 μm nozzle) is performed on a Sintech 20 G (MTS, USA) with a 100 kN load cell. All experiments are performed at an initial strain rate of $5 \times 10^{-4}$ s$^{-1}$. Specimens for microscopic cross-section analysis are embedded into epoxy mounting resin (Epothin2, Buehler, USA), ground with SiC grinding paper (P600 to P2500), polished using 6, 3, and 1 μm diamond suspensions, and lapped with colloidal silica. Scanning electron microscopy is performed on a FEI Quanta 650 equipped with an Oxford EDX detector on carbon-coated, lapped specimens. Oxide powder specimens are coated with 8 nm Os. Porosity is measured using thresholding in ImageJ. Linear shrinkage is calculated from longitudinal filament size measurements taken prior and after sintering. 2D X-ray diffraction on milled $Co_3O_4 + Cr_2O_3 + Fe_2O_3 + NiO$ feedstock powder prior and after TGA is performed on a Bruker KAPPA APEX II with a Mo source.

## Data availability

The data that support the findings of this study are available from the corresponding author upon reasonable request.

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

## Acknowledgements

C.K. received funding from the Swiss National Science Foundation as an Early Postdoc Mobility fellowship under grant No. 172180. The authors thank the Paul Scherrer Institut, Villigen, Switzerland for the provision of beamtime at the X04SA beamline of the Swiss Light Source, Dr. A. Pinar for providing the gas capillary system, and M. Lange for technical support. We gratefully acknowledge Prof. R. Shah for useful discussion and access to her Bioplotter for 3D printing and Prof. S. Haile and Dr. T. Davenport for the TGA measurements. This work made use of the EPIC facility of Northwestern University's NUANCE Center and the IMSERC, which have received support from the Soft and Hybrid Nanotechnology Experimental (SHyNE) Resource (NSF ECCS-1542205); the State of Illinois and International Institute for Nanotechnology (IIN). This work made use of the Central Laboratory for Materials Mechanical Properties and the MatCI Facility that together with EPIC received support from the MRSEC program (NSF DMR-1720139) at the Materials Research Center.

## Author contributions

C.K. and D.C.D. conceived the project. N.P.M.C. prepared, and together with C.K. performed, the in situ XRD experiments. C.K. performed all other experiments and evaluated the data. All authors discussed the findings and contributed to writing of the manuscript.
