## [Peer Review File · Nature Communications]

Reviewers' comments:

Reviewer #1 (Remarks to the Author):

In this manuscript, the authors developed a two-step method to fabricate the CoCrFeNi high-entropy-alloy (HEA) filaments and 3D microlattices. This two-step method involves the 3D extrusion of inks and subsequent sintering at high temperature and in H₂ atmosphere. Such method can be used to synthesized the CoCrFeNi filament with diameter of about 100 microns. The authors also investigated the influence of fabrication processing (such as temperature, nozzle diameter etc.) on the structural changes of fabricated filaments. Overall, some results are interesting and the manuscript is well-written. The remarkable novelty of current study is that the diameter of filament prepared by the current two-step method is smaller than that of other 3D printing methods for metallic materials. But there are a series of following aspects which need to be further addressed, before the manuscript can be recommended for publication.

1. In the current 3D ink printing, the authors used the powders of various oxide materials. It is well-known the powder size and shape have a certain influence on the printed structures. Therefore, the authors are strongly suggested to provide the information of powder size and shape of various oxide materials used in the current study, and to further add the possible discussion or analyses on their influences.

2. In the bottom of Fig. 3, the authors showed that the grain size of fabricated filament is 5-25 micron after sintering for 20 min at 1573 K. On the paragraph of page 9, the authors tested the fully sintered filament with grain size of 35-80 micron. There is a distinct difference in the grain size between these two filaments. Whether such difference is related to different processing conditions? The authors should clarify this. Moreover, it is well-known that the metals/alloys with the smaller grain size have the higher strength and the better other properties. Why not the authors fabricated the filaments with smaller grain size under 20 min sintering to achieve the higher strength or better properties?

3. In the current study, the authors printed the cross-ply microlattices and preformed compressive tests on these samples. In the manuscript, the authors only reported one tested case. How many samples did the authors prepare? How many compressive tests did the authors conduct? The authors should provide these information and show all the stress-strain curves of all tested microlattices. Furthermore, the mechanical properties and deformation of microlattice is also related to the geometry of unit cell. The printed microlattices in the current study have a simple geometry (cross-ply). Can the current method be used to print more complicated geometries (for example, octet-, tetra- and iso-trusses)?

4. Most recent study (Nano Lett., 2018, 18, 4247-4256) reported the fabrication and mechanical testing of 3D HEA-polymer composite nanolattices with octet-truss unit cells. These nanolattices are made up of a high-entropy alloy-coated (14.2–126.1 nm in thickness) polymer strut (approximately 260 nm in the characteristic size) fabricated via two-step process, involving 3D printing based on two-photon lithography and magnetron sputtering deposition. These nanolattices have both high strength and good recoverability, overcoming the strength-recoverability tradeoff. The authors should review this study and compare the current study with this study about HEA nanolattice. In the current study, the authors only tested the compression of microlattices. The authors are suggested to further investigate the recoverability of microlattices by unloading the overall structure before its fracture. Moreover, the CoCrFeNi HEA has good mechanical properties at the cryogenic temperature. Therefore, the reviewer suggested that the authors should conduct the mechanical testing of both HEA filaments and microlattices at the cryogenic temperature to show the mechanical properties of fabricated filaments and microlattices.

5. The current fabrication method can be used to the CoCrFeNi HEA system. Previous studied have shown that the CoCrFeNi HEA can be served as a base alloy and designed lots of new alloys by adding alloying elements. Can such method be applicable to other HEA systems?

6. On page 9, the authors compared the tensile strength and the fracture strain of fabricated HEA filaments with those from HEA wires fabricated previously by the conventional alloying methods. The mechanical properties of materials are mainly determined by their microstructures and fabrication processing. Except mechanical properties, the authors are strongly suggested to make

a table to compare the microstructures (such as grain size, phase, lattice constant) and processing of current samples with previous samples.

7. The HEA microlattices exhibited the extensive plastic and densification regimes. The authors are suggested to show some typical snapshots of compressed microlattice to reveal the deformation details.

8. Recent perspective (Nature Mater., 2016, 15, 373-374) indicated the smaller and the stronger for mechanical metamaterials. In the current study, the authors can use the nozzle with different diameter to control the diameter of filament. Did the authors try the nozzle with smaller diameters or other methods to further reduce the diameter of filament?

Reviewer #2 (Remarks to the Author):

The paper presents an experimentally thorough investigation of HEA's. The method used to make the alloys is quite novel, and from this perspective, I very much enjoyed reading the paper. The XRD and microscopy work are both excellent, and I only have a few comments for the authors, and I hope they find these comments constructive.

The authors might consider an SEM EDS map of the extruded filament shown in the top and middle centre micrographs of Fig 3. At present, identification of what chemical composition correlates with the grey scale SEM images is not conclusive (mainly because oxygen concentration can modify the global average atomic number, thereby affecting the backscattered electron yield vis-à-vis grey scale). I think one good chemical map would tighten this aspect of the paper.

Porosity is notoriously difficult to measure. Particle pull-out and pore etching/cavitation during metallographic polishing are both well known to artificially inflate the porosity value. The model of pore formation shown in fig 4b is not convincing. The linear shrinkage and porosity don't seem to correlate as they should either. Measuring 1, 2 or 3% porosity via the method described is basically the same porosity level within experimental error, and highly dependant on thresholding levels. Of most importance, the porosity is inhomogenous, so the values measured by local observation will naturally have a large error bar. I have two suggestions for the authors to consider. One option for them is to measure the density of the samples if they have access to a high sensitivity machine. This should fix the measurement issues mentioned here in regards to porosity measurement. Alternatively, in the present paper, the porosity is really just a side note to the work, the authors may consider shortening the porosity discussion, and simply state that the porosity decreased with temperature and time, and it plateaued out at 1-4%.

The changes in flow stress resulting from changes to filament length are perplexing. Usually, the longer the wire, the higher chance of the sample containing a critical flaw. Thus, longer fibres usually exhibit lower average strength values. The best reference I could find was for carbon fibres, but I suspect the concept is universal [Moreton, R. The effect of gauge length on the tensile strength of R.A.E. carbon fibres. Fibre Sci. Technol. 1, 273-284 (1969)]. This is diametrically opposite to what you find. Again I advise the authors to consider brevity as a positive attribute, and perhaps remove this section. The authors could simply report the 10 mm length results, and state in one line that smaller samples showed higher variability. This higher variability with smaller length fibres may be an experimental problem (possibly inadvertent damage from handling such small samples, or perhaps the short samples all came from the one print, and the longer ones from another batch???)

With regards to the size of the wire, it is established that one needs ~12 grains across a gage length to provide a measured flow stress indicative of the bulk properties. In your case you have less, and according to [Hansen Acta Mat. vol 25 (1977) and Armstrong J. Mech Phys. Solids 9, 196 (1961)] for a given grain size, the measured "strength decreases with decreasing specimen diameter due to a reduction in constraint strengthening". Thus in your case you measure a smaller

flow stress than you would have in a bulk specimen of the same microstructure. This may be worth some mention in the text.

I apologize for this rather long review which really only has four relatively minor queries for the authors to consider. I enjoyed the paper, it was refreshingly different to others in this field.

Reviewer #3 (Remarks to the Author):

The article deals with the preparation of an HEA 3D lattice by 3D-inkjet printing. The HEA is formed by hydrogen reduction of nanometer sized oxides+ after debinding. The reduction of Chromium oxide is realised by highly pure hydrogen at 1573 K. The article is well written, clearly structured and may well influence and stimulate research in the field of additive manufacturing of HEAs.

Comments:

page 4 line 100 (figure 1 caption): without warping of cracking
replace by without warping or cracking

page 8 line 182

Larger agglomerates present undergo local sintering to form spongy Co and Ni
Spongy Ni and Co may as well arise from the volume reduction due to the chemical reduction.
From the hydrogen reduction in iron oxide it is known that the surface area of the formed metal can increase drastically leading to pyrophoric iron (e.g. Helvetica Chimica Acta Vol. 47/1, 1964).

page 8 208-216 (void formation)

A different mechanism is supposed. From Sintering Al-Ti-blends it is known that Al diffuses into the surrounding Ti particles causing those to expand and leaving large pores in its original place. In the observed case chromium would diffuse into the HEA causing its lattice to expand (see XRD) and additionally the volume increases due to the increasing amount of metallic material, which would also explain the observed swelling (see

Bohm, A; Kieback, B ZEITSCHRIFT FUR METALLKUNDE Volume: 89 Issue: 2 Pages: 90-95
Published: FEB 1998).

general remark

It is highly surprising that chromium oxide can be reduced once it is fully encapsulated by a metallic alloy. If there is no confirmation of a full encapsulation, the hypothesis about H₂/H₂O filled pores is not supported and should be omitted.

page 10 line 283: pack aluminization might lead to phase formation which may hamper the mechanical properties of the alloy. However this is only in the outlook section, anyway.

page 11 line 299: if possible, please include the dew point of the hydrogen.

Dear Reviewers,

Thank you for your comments, input and suggestions, which were very useful to improve our manuscript. Each comment is individually addressed in the following section.

Reviewer 1:

1. In the current 3D ink printing, the authors used the powders of various oxide materials. It is well-known the powder size and shape have a certain influence on the printed structures. Therefore, the authors are strongly suggested to provide the information of powder size and shape of various oxide materials used in the current study, and to further add the possible discussion or analyses on their influences.

We agree and **fully disclose powder information** in the Methods section, allowing easy replication by readers: "The extrudable ink (target composition after co-reduction: 25Co-25Cr-25Fe-25Ni, at.%) with 70:30 vol.% powder-to-polymer ratio is prepared from Fe₂O₃ (2.57 g, ≥99%, <5 μm, Sigma-Aldrich), NiO (2.40 g, 99%, -325 mesh, Sigma-Aldrich), Co₃O₄ (2.58 g, <10 μm, Sigma-Aldrich) and Cr₂O₃ (2.45 g, 99.7%, -325 mesh, Alfa Aesar) powders, poly-lactic-co-glycolic-acid (PLGA, 0.87 g, 82:18, Evonik Industries) as binder, dibutyl phthalate (DBP, 1.58 g, Sigma-Aldrich) as plasticizer, ethylene glycol butyl ether (EGBE, 0.79 g, Sigma-Aldrich) as surfactant and methylene chloride (20 ml, DCM, Sigma-Aldrich) as solvent.

The resulting powder size after the mill/mixing of the ink is also shown in Figure 1b, demonstrating a particle size <1 μm and agglomerate sizes <10 μm enabling smooth flow of the ink.

2. In the bottom of Fig. 3, the authors showed that the grain size of fabricated filament is 5-25 micron after sintering for 20 min at 1573 K. On the paragraph of page 9, the authors tested the fully sintered filament with grain size of 35-80 micron. There is a distinct difference in the grain size between these two filaments. Whether such difference is related to different processing conditions? The authors should clarify this. Moreover, it is well-known that the metals/alloys with the smaller grain size have the higher strength and the better other properties. Why not the authors fabricated the filaments with smaller grain size under 20 min sintering to achieve the higher strength or better properties?

Thank you for bringing up this interesting point. The difference in grain size is due to the increased sintering time to obtain higher density. After **20min** at 1573 K indications for swelling and potentially higher porosity were found. Thus all mechanical testing is performed on samples sintered at 1573 K for **1 h**, where the lowest amount of porosity was observed. Despite a potential benefit due to smaller grain size, we decided to test the conditions with the highest material integrity.

We added clarification that tested filaments were sintered at 1573K/1h.

“Fully sintered filaments (1573K, 1h), with a nearly circular cross-section and a grain size of 35-80 μm ,”

3. In the current study, the authors printed the cross-ply microlattices and performed compressive tests on these samples. In the manuscript, the authors only reported one tested case. How many samples did the authors prepare? How many compressive tests did the authors conduct? The authors should provide these information and show all the stress-strain curves of all tested microlattices. Furthermore, the mechanical properties and deformation of microlattice is also related to the geometry of unit cell. The printed microlattices in the current study have a simple geometry (cross-ply). Can the current method be used to print more complicated geometries (for example, octet-, tetra- and iso-trusses)?

Originally, one lattice was tested and hence also reported. Based on comments from the other reviewers, **a new series of four lattices was produced, sintered and tested** to demonstrate the mechanical properties at cryogenic temperatures. Data of all 5 lattices is shown now in Figure 4.

We agree that many other geometries can be printed, with the caveat that this is an extrusion process, limiting the amount of overhang that can be produced without support. We use a simple structure here to demonstrate the potential of the 3D ink-extrusion process of blended oxides to produce CoCrFeNi alloy with excellent properties after reduction/sintering. We added: “Additionally, **this method can create many other printed micro-lattice geometries** and the 3D ink-extruded structures can be modified in their green body state, as demonstrated by creating origami and kirigami structures from ink-extruded TiH₂ sheets.”

4. Most recent study (Nano Lett., 2018, 18, 4247-4256) reported the fabrication and mechanical testing of 3D HEA-polymer composite nanolattices with octet-truss unit cells. These nanolattices are made up of a high-entropy alloy-coated (14.2–126.1 nm in thickness) polymer strut (approximately 260 nm in the characteristic size) fabricated via two-step process, involving 3D printing based on two-photon lithography and magnetron sputtering deposition. These nanolattices have both high strength and good recoverability, overcoming the strength-recoverability tradeoff. The authors should review this study and compare the current study with this study about HEA nanolattice. In the current study, the authors only tested the compression of microlattices. The authors are suggested to further investigate the recoverability of microlattices by unloading the overall structure before its fracture. Moreover, the CoCrFeNi HEA has good mechanical properties at the cryogenic temperature.

We agree that the work on 3D printed polymer scaffolds coated with HEA alloy, thus creating a composite structure, is an interesting approach to HEA nanolattices. **It has been added in the introduction** : “... and polymer-CoCrFeNi composites by coating 260 nm thick 3D printed polymer nanolattices with 14-126 nm thick HEA shells.”

The structures created in our work are however ~3 orders of magnitude larger (100s of μm vs. 100 nm) and thus we do not observe size effects related to minimizing the alloy into the nano-scale. Another difference is that our lattices are fully metallic (and thus capable of being sintered at 1573K, compared to a polymer-HEA composite), and plastically deform under compression. Apart from a small elastic portion, deformation is permanent and not recoverable, unlike in elastic polymer-HEA nanolattices where most of the material is polymer and the metal is only a thin shell, and similar to a typical metal tested under compression.

Therefore, the reviewer suggested that the authors should conduct the mechanical testing of both HEA filaments and microlattices at the cryogenic temperature to show the mechanical properties of fabricated filaments and microlattices.

Based on this comment, **we created a new series of specimens and performed tensile testing of single sintered HEA filaments at 130 K and compression testing of a new series of 0/90° cross-ply lattices at 77 K demonstrating the excellent properties of the alloy after 3D ink-extrusion, co-reduction and sintering.** This has been added to Figure 4.

5. The current fabrication method can be used to the CoCrFeNi HEA system. Previous studies have shown that the CoCrFeNi HEA can be served as a base alloy and designed lots of new alloys by adding alloying elements. Can such method be applicable to other HEA systems?

This is a very relevant question, to which the answer is affirmative. If co-reduction of oxides is used, the restriction is the ability to reduce in a H_2 atmosphere. But many transition metals can indeed be reduced to their metallic form in H_2 , e.g. Fe, Ni, Mn, Cr, Cu, Zn, Mo, W, Ag. If hydrides are used, whose decomposition creates the metal, other transition elements become possible, such as Ti, Zr, Hf, V, Nb, Ta. Also, blends of elemental powders can be used, removing the reduction step and simply necessitating interdiffusion. The 3D ink-extrusion approach is independent of powder chemistry and thus can be flexibly adapted. **In the manuscript this is included** as: "The use of blended powders - oxide as shown here, but which could be also metallic - allows full compositional flexibility well beyond the particular CoCrFeNi composition studied here, and even compositionally-graded HEAs which can be printed by changing the mixing ratios of multiple inks. This will also allow this technique to be used in alloy development by creating a variety of compositions, in short times, for combinatorial discovery of novel alloys, as demonstrated using beam-based deposition of metal powder blends."

Typical compositions that could be envisaged would be CoCrFeMnNi by co-reduction of five oxides, AlCoCrFeNi by addition of an Al-source, TiZrHfV from hydride decomposition and a complex ZrTiVCrFeNi from elemental powder or a mixture of elemental powders (Fe, Ni, Cr) and hydrides (Zr, Ti, V).

6. On page 9, the authors compared the tensile strength and the fracture strain of fabricated HEA filaments with those from HEA wires fabricated previously by the conventional alloying methods. The mechanical properties of materials are mainly determined by their microstructures and fabrication processing. Except mechanical properties, the authors are strongly suggested to make a table to compare the microstructures (such as grain size, phase, lattice constant) and processing of current samples with previous samples.

We agree and have **provided a Supplementary Table**, compiled from the scarce information available. Unfortunately very limited microstructural information is provided by the authors cited.

Method / Geometry	Specimen diameter (mm)	Yield stress (MPa)	Ultimate tensile strength (MPa)	Elongation to failure (%)	Grain size (μm)	Reference
Cold-drawn wire	7	1107	1107	12.6	4.1	
Cold drawn + recrystallized wire	7	288	NA	NA	NA	Huo et al. ⁴⁶
Arc melted, dog-bone	1x0.5	200	370	41	200-300	Huo et al. ⁴⁷
Arc melted, dog-bone	6	229	454	36	NA	Huang et al. ⁴⁸
Printed, reduced from oxides, sintered filament	0.125	250 \pm 5	598 \pm 8	33.8 \pm 1.3	35-80	This work

7. The HEA microlattices exhibited the extensive plastic and densification regimes. The authors are suggested to show some typical snapshots of compressed microlattice to reveal the deformation details.

This is a good suggestion, and **images of uncompressed and a compressed lattices (-50%, 77 K) are added to Figure 4. An animated GIF of the compression process at 293 K is uploaded as additional supplementary file.** SEM-level details as in Nano Lett., 2018, 18, 4247-4256 cannot be provided as mechanical testing of these lattices is performed on a 100 kN load frame. Also for 77 K testing submerged in LN2, no imaging data is available.

8. Recent perspective (Nature Mater., 2016, 15, 373-374) indicated the smaller and the stronger for mechanical metamaterials. In the current study, the authors can use the nozzle with different diameter to control the diameter of filament. Did the authors try the nozzle with smaller diameters or other methods to further reduce the diameter of filament?

Yes, different nozzles were used with diameters of 200, 250, 330 and 510 μm . Nozzle sizes below 200 μm are not explored as the Bioplotter uses compressed air at a maximum pressure of 5 bar to drive the piston in the extrusion barrel. This, together with the agglomerate size in the ink, limits the minimum nozzle diameter that can be used. However, binder burn-out and shrinkage upon co-reduction and sintering reduce

the filament diameter from 200 μm to 103 μm . In this sense the use of an oxide-based ink is an approach to obtain filaments below the extrusion nozzle diameter.

Reviewer 2:

1. The authors might consider an SEM EDS map of the extruded filament shown in the top and middle centre micrographs of Fig 3. At present, identification of what chemical composition correlates with the grey scale SEM images is not conclusive (mainly because oxygen concentration can modify the global average atomic number, thereby affecting the backscattered electron yield vis-à-vis grey scale). I think one good chemical map would tighten this aspect of the paper.

We agree, and we have added an EDS Map of the center specimen combining metallic matrix and oxide particles (Supplementary Figure S3). Oxygen and Cr are co-located as well as Co, Fe, Ni and Cr in the metallic matrix.

2. Porosity is notoriously difficult to measure. Particle pull-out and pore etching/cavitation during metallographic polishing are both well known to artificially inflate the porosity value. The model of pore formation shown in fig 4b is not convincing. The linear shrinkage and porosity don't seem to correlate as they should either. Measuring 1, 2 or 3% porosity via the method described is basically the same porosity level within experimental error, and highly dependant on thresholding levels. Of most importance, the porosity is inhomogenous, so the values measured by local observation will naturally have a large error bar. I have two suggestions for the authors to consider. One option for them is to measure the density of the samples if they have access to a high sensitivity machine. This should fix the measurement issues mentioned here in regards to porosity measurement. Alternatively, in the present paper, the porosity is really just a side note to the work, the authors may consider shortening the porosity discussion, and simply state that the porosity decreased with temperature and time, and it plateaued out at 1-4%.

We do not have access to an instrument of the required precision to measure porosity in such small fibers. To address the above issue, we have shortened the porosity discussion, as suggested by the reviewer.

3. The changes in flow stress resulting from changes to filament length are perplexing. Usually, the longer the wire, the higher chance of the sample containing a critical flaw. Thus, longer fibres usually exhibit lower average strength values. The best reference I could find was for carbon fibres, but I suspect the concept is universal [Moreton, R. The effect of gauge length on the tensile strength of R.A.E. carbon fibres. Fibre Sci. Technol. 1, 273-284 (1969)]. This is diametrically opposite to what you find . Again I advise the authors to consider brevity as a positive attribute, and perhaps remove this section. The authors could simply report the 10 mm length results, and state in one line that smaller samples showed higher variability. This higher variability with smaller length fibres may be an experimental problem (possibly inadvertent damage from handling such small samples, or perhaps the short samples all came from the one print, and the longer ones

from another batch???

We indeed have found that longer fibers have lower strength and fail at lower strain, in agreement with the reviewers comment "the longer the wire, the higher chance of the sample containing a critical flaw". All specimens for the room temperature tensile tests are from the same ink batch and extrusion job. The 3 specimens at 10 mm and 3 of the 1 mm were sintered together, the other 3 at 1 mm were sintered in the same furnace under the same conditions in another batch. Fibers were selected randomly out of the sintered fibers for testing at 10 mm or 1 mm gauge length.

In the main paper only the 10 mm gauge length results are used, also to compare against testing at cryogenic 130 K, based on Reviewer 1's comments.

4. With regards to the size of the wire, it is established that one needs ~12 grains across a gage length to provide a measured flow stress indicative of the bulk properties. In your case you have less, and according to [Hansen Acta Mat. vol 25 (1977) and Armstrong J. Mech Phys. Solids 9, 196 (1961)] for a given grain size, the measured "strength decreases with decreasing specimen diameter due to a reduction in constraint strengthening". Thus in your case you measure a smaller flow stress than you would have in a bulk specimen of the same microstructure. This may be worth some mention in the text.

We agree and **we added to the manuscript:**

In general, the measured yield stress in this work is expected to be lower compared to the bulk value due to the small number of grains across the filament diameter thus reduced constraint strengthening.

Reviewer 3:

1. page 4 line 100 (figure 1 caption):without warping of cracking, replace by without warping or cracking
Thanks for catching this typo, we have fixed it.
2. page 8 line 182
Larger agglomerates present undergo local sintering to form spongy Co and Ni
Spongy Ni and Co may as well arise from the volume reduction due to the chemical reduction. From the hydrogen reduction in iron oxide it is known that the surface area of the formed metal can increase drastically leading to pyrophoric iron (e.g. Helvetica Chimica Acta Vol. 47/1, 1964).
Agreed, and text adapted:
Larger agglomerates, undergoing local sintering as well as volume reduction of larger oxide particles via chemical reduction, lead to the formation of porous, spongy Co and Ni.
3. page 8 208-216 (void formation)

A different mechanism is supposed. From Sintering Al-Ti-blends it is known that Al diffuses into the surrounding Ti particles causing those to expand and leaving large pores in its original place. In the observed case chromium would diffuse into the HEA causing its lattice to expand (see XRD) and additionally the volume increases due to the increasing amount of metallic material, which would also explain the observed swelling (see Bohm, A; Kieback, B ZEITSCHRIFT FUR METALLKUNDE Volume: 89 Issue: 2 Pages: 90-95 Published: FEB 1998).

We agree, and the text has been adapted for the suggested mechanism.

The slight isothermal swelling phenomenon observed at 1573 K is attributed to diffusion of freshly reduced Cr into the surrounding metallic matrix, thereby expanding its lattice parameter, as seen by XRD, and the overall volume³¹ (Figure 4b).

4. It is highly surprising that chromium oxide can be reduced once it is fully encapsulated by a metallic alloy. If there is no confirmation of a full encapsulation, the hypothesis about H₂/H₂O filled pores is not supported and should be omitted.

We agree and have removed this hypothesis.

5. page 10 line 283: pack aluminization might lead to phase formation which may hamper the mechanical properties of the alloy. However this is only in the outlook section, anyway.

This is a good point, and might be the topic of a future study; however based on results on other alloys systems (e.g., pack aluminization of Ni, Ni-Cr foams) we are confident that conditions exist where the aluminides on the surface can be successfully dissolved and the composition can be homogenized.

6. page 11 line 299: if possible, please include the dew point of the hydrogen.

The used hydrogen has up to 2 ppm (volume) of H₂O. The calculated dew point would be -72 °C or 201 K. This information is added.

REVIEWERS' COMMENTS:

Reviewer #1 (Remarks to the Author):

After reading the revised manuscript and the responses to three reviewers, I felt that the authors have addressed most concerns from three reviewers and that the manuscript has been improved much. There is one typo in the revised manuscript, i.e. the composition of HEA in Ref. 12 should be AlCoCrFeNi not CoCrFeNi. Moreover, the authors are suggested to add a discussion or outlook on "the smaller the stronger" of HEA microlattices according to the perspective (Nature Mater., 2016, 15, 373-374), which might stimulate more attention and more extensive studies. Once the authors finish these revisions, the paper will be recommended for publication without further review.

Reviewer #2 (Remarks to the Author):

I am satisfied that the authors have addressed all of the concerns that were raised.

Dear Reviewers,

Thank you for your comments, input and suggestions, which were very useful to improve our manuscript. Each comment is individually addressed in the following section.

Reviewer 1:

1. There is one typo in the revised manuscript, i.e. the composition of HEA in Ref. 12 should be AlCoCrFeNi not CoCrFeNi.

Thank you for catching this typo, **it has been changed to AlCoCrFeNi.**

2. Moreover, the authors are suggested to add a discussion or outlook on “the smaller the stronger” of HEA microlattices according to the perspective (Nature Mater., 2016, 15, 373-374), which might stimulate more attention and more extensive studies.

The following discussion/outlook has been added:

With recent advances in AM in the micron scale using lithography and direct laser writing techniques, sub-micron scale lattices forming metamaterials and HEA-polymer composites demonstrating superior mechanical properties have been demonstrated. Thus miniaturization of HEA cellular structures provides a potential path to discovery of combinations of structures and alloys further pushing the limits of their mechanical performance beyond the – already impressive – bulk properties of HEAs. The approach presented in this work, co-reduction of blended oxide nanopowders, can be envisaged as a method to add a variety of HEA alloys to cellular preforms produced by AM to create complex composites or purely metallic HEA micro- and nano-lattices.